# An Improved Mask R-CNN Micro-Crack Detection Model for the Surface of Metal Structural Parts

**DOI:** 10.3390/s24010062

**Published:** 2023-12-22

**Authors:** Fan Yang, Junzhou Huo, Zhang Cheng, Hao Chen, Yiting Shi

**Affiliations:** School of Mechanical Engineering, Dalian University of Technology, Dalian 116024, China; youngfine@mail.dlut.edu.cn (F.Y.); chengzhang@mail.dlut.edu.cn (Z.C.); chenhap@mail.dlut.edu.cn (H.C.); ytshi8088@163.com (Y.S.)

**Keywords:** mask R-CNN, micro-crack, target detection, metal structural parts, deformable convolution kernel, attention mechanism

## Abstract

Micro-crack detection is an essential task in critical equipment health monitoring. Accurate and timely detection of micro-cracks can ensure the healthy and stable service of equipment. Aiming at improving the low accuracy of the conventional target detection model during the task of detecting micro-cracks on the surface of metal structural parts, this paper built a micro-cracks dataset and explored a detection performance optimization method based on Mask R-CNN. Firstly, we improved the original FPN structure, adding a bottom-up feature fusion path to enhance the information utilization rate of the underlying feature layer. Secondly, we added the methods of deformable convolution kernel and attention mechanism to ResNet, which can improve the efficiency of feature extraction. Lastly, we modified the original loss function to optimize the network training effect and model convergence rate. The ablation comparison experiments shows that all the improvement schemes proposed in this paper have improved the performance of the original Mask R-CNN. The integration of all the improvement schemes can produce the most significant performance improvement effects in recognition, classification, and positioning simultaneously, thus proving the rationality and feasibility of the improved scheme in this paper.

## 1. Introduction

In practical engineering applications, the service life of metal structural components usually directly determines whether engineering equipment can reliably, stably, and efficiently complete construction tasks. Since the emergence and expansion of micro-cracks typically harm the metal materials’ strength, indirectly shortening the service life of critical structural components and even the equipment itself, it is necessary to identify the early emergence of micro-cracks and repair or replace the affected parts during the engineering equipment service process to reduce the economic losses caused by fatigue failure. Therefore, the efficient and accurate identification and detection of micro-cracks on the surface of metal structural components has significant application potential and research value.

In the earlier days, the artificial visual method was mainly used to detect the micro-cracks on the metal structural components’ surface. Still, limited by a series of factors such as the technology of the inspectors and the lighting conditions at the site, it is difficult to ensure the accuracy and speed of the inspection using the artificial visual inspection method. Subsequently, many scholars proposed a variety of sensor-based micro-crack detection methods, such as ultrasonic, eddy current, acoustic emission, etc. However, there are certain limitations, such as the object material, surface topography, and other factors, resulting in a general lack of flexibility of the sensor-based crack detection methods, and making it challenging to improve the detection accuracy, speed, and adaptability.

Compared to various sensors, using machine vision methods to acquire crack images and perform a series of processing tasks can circumvent some constraints. The core of traditional machine vision-based algorithms for small target detection lies in using feature extractors for image feature extraction, which includes three main steps: selecting candidate target regions, extracting features, and designing target classifiers. Many scholars have proposed corresponding optimization methods built around the above three steps, such as more flexible sliding windows [1], an extraction method based on multi-scale features [2], and the better performance classification method XGBoost [3] (XGBoost is a scalable machine learning system for tree boosting and is an improved version of GBDT (Gradient Boosting Decision Tree), where X stands for eXtreme. XGBoost is a widely used machine learning method, which can train models faster and more efficiently). However, even after nearly 20 years of development, traditional methods still have specific weaknesses, such as using sliding windows to select the target region producing more redundant windows, resulting in the overall process being complex; as such the sliding window scale flexibility, limitations, and robustness are poor. In addition, the features based on low-level visual information are difficult to adapt to complex and changeable scenes, which ultimately means that the efficiency and accuracy of the traditional method in dealing with the small targets detection task can be little further improved, and as such the promotion and application of the method is restricted.

Several researchers have combined machine vision with traditional analytical methods to investigate the detection of micro-cracks. Landstrom et al. [4] proposed an automated online crack detection method based on the crack detection task on steel plate surfaces using morphological image processing and statistical classification based on logistic regression, which successfully identified more than 70% of the manually labeled crack lengths, missing only a few crack regions that contained shorter crack segments. Cubero-Fernandez et al. [5] proposed a single-step crack detection and classification method based on algorithms such as logarithmic transformation, bilateral filter, Canny operator, and morphological filtering to achieve an automated pavement crack detection system free from manual operation.

In recent years, with the rapid improvement in the performance of CPUs, GPUs, and other computing units, as well as camera CMOSs, optical imaging lenses, and other imaging hardware, both the field of computer vision and that of deep learning have witnessed rapid development. The traditional target detection task has gained a new direction and ideas for exploration.

Krizhevsky [6] proposed the first deep convolutional neural network, AlexNet, using ImageNet as a training dataset, which achieved breakthrough detection results on the ImageNet Large Scale Visual Recognition Challenge (ILSVRC). The successful application of AlexNet in classification tasks became a starting point, and since then, deep learning has been widely applied and developed in other visual tasks.

Target detection technology based on deep learning can use the multi-structured network model and its powerful training algorithm to adaptively learn the images’ high-level semantic information. After extracting the image features and inputting them to the classification network to complete the classification and localization task of the target, the method effectively improves the accuracy and efficiency of the target detection, so target detection technology based on deep learning has become a hotspot of attention for many research scholars.

According to whether there is a candidate box generation step in the target detection process, the target detection algorithms can be divided into two-stage and single-stage detection algorithms, where the steps of the two-stage detection algorithm include (1) generating candidate regions and (2) determining candidate region categories and fine-tuning the bounding box. The single-stage detection algorithm discards the candidate region generation stage and directly defaults to all locations on the image as potential candidate regions and attempts to categorize each region of interest as either a background or target object.

Through the continuous research of scholars, there have been many excellent models of target detection algorithms. Two-stage algorithms include Fast R-CNN [7], Faster R-CNN [8], SPPNet [9], R-FCN [10], etc., whose distinctive feature is that the detection accuracy is higher, but the speed is slightly slower, which makes it difficult to deal with detection tasks that have a high real-time requirement. Single-stage detection algorithms include the YOLO [11] series, SSD [12], DSSD [13], RetinaNet [14], RefineNet [15], etc., whose significant feature is that the detection speed is very high, but the detection accuracy is slightly insufficient compared to two-stage detection algorithms.

Although the above algorithms can identify and classify targets with high efficiency, they cannot distinguish different individuals in the same class of targets. Mask R-CNN, proposed by He et al. [16], is based on Faster R-CNN extended with a branch for instance mask prediction, which is parallel to the existing bracketed frame regression branch and classification branch and thus can simultaneously perform target detection and instance segmentation. Furthermore, it adds instance segmentation loss in the loss function, so Mask R-CNN has faster detection speed and good detection effect in the field of pose estimation and character key point detection. Mask R-CNN is one of the most commonly used image instance segmentation algorithms, and it is widely used in defective target segmentation tasks, such as road surfaces [17], industrial manufacturing [18], bolt fasteners [19], leather surfaces [20], etc. However, the above studies mainly focus on the target detection tasks with closer horizontal and vertical dimensions and more prominent features, while the model still struggles with target detection tasks such as micro-cracks, which have significant differences in horizontal and vertical dimensions and features that are easily lost.

In recent years, many scholars have carried out various studies in order to improve the recognition performance of Mask R-CNN. Zhou et al. [21] proposed an improved Mask R-CNN model based on attention, rotation, and genetic algorithm for defect detection of damaged insulators in power equipment. The model’s sensitivity to small targets can be significantly improved by modifying the backbone, and the location of small targets can be identified more quickly. Meanwhile, the genetic algorithm and gradient descent algorithm are combined to optimize the model’s hyperparameters. The model training results are as close to the global optimal as possible. Shen et al. [22] proposed an unsound wheat kernel recognition algorithm based on improved Mask R-CNN for the needs of rapid wheat rating. By optimizing the structure of FPN and RPN, and adding attention mechanism, the model can identify unsound wheat kernels faster and more accurately, with an accuracy rate of 86%, a recall rate of 91%, and the inference speed of a single image reaching 7.83 s. Aiming at the problems of low efficiency and low accuracy during the manual maintenance of railway switches, Wei et al. [23] proposed a low-complexity accurate ranging algorithm based on Mask R-CNN. The region of interest is segmented twice through the interactive iterative method, the image distortion is corrected according to the vertex mapping principle, and the accurate actual distance can be calculated by fitting the linear distance transformation equation. Finally, it can achieve accurate calculation of small targets and accurately measure the distance between different working parts.

At the same time, Mask R-CNN can be well applied to the identification, detection, and analysis of specific small target objects. Aiming at the statistical requirements of determining the number of air conditioners in use, Yang et al. [24] established the feature detection dataset of external air conditioners, proposed the automatic search algorithm of urban air conditioners based on Mask R-CNN and YOLOv5, and explored the feasibility of using street view images to identify the number of air conditioners. Aiming at the demand for sows’ real-time detection and recognition in the process of large-scale sow breeding, Lei et al. [25] established a multi-objective sow detection and recognition model based on improved Mask R-CNN and UNet-Attention deep learning algorithms, which could identify the contour of sows and analyze the area distribution of sows in the pen. At the same time, it can also recognize the behavior of the sow, such as eating, fetching water, and lying down, and the final recognition rate reached 96.8%.

In recent years, some scholars have also explored new methods for crack detection. Wang et al. [26] presented a SIFT matching method based on an alternate-selection strategy, solved the problems of local information losses and partial refinement capacity reductions which are frequently encountered in the crack detection algorithms of deep learning methods. Luo et al. [27] reviewed three significant aspects of CV-based methods, including surface defect detection, vibration measurement, and vehicle parameter identification, which aimed to provide guidance for selecting appropriate CV-based methods for bridge inspection and monitoring. Computer vision and deep learning methods give crack detection more space to explore.

By researching traditional and deep learning-based target detection methods, it can be found that although the current mainstream target detection algorithms have been able to achieve a better balance between target detection speed and accuracy, there are still specific problems in the presence of tiny targets, such as the lack of high-quality open datasets of tiny target defect samples, the difficulty of accurately extracting tiny target features, features that tend to be submerged in the complex background and texture, and tiny target detection accuracy that still has space for improvement. Therefore, this paper focuses on the above difficulties, takes micro-cracks on the surface of metal structural parts as the research object, collects micro-crack images in advance to form a certain scale of micro-crack dataset, and carries out data enhancement, at the same time conducting optimization research on micro-crack detection algorithms based on the deep learning method, to provide methodological support for improving the accuracy of detecting micro-cracks on the surface of metal structural parts under different working conditions.

## 2. Related Work

### 2.1. Workflow of Mask R-CNN Network

Mask R-CNN is mainly composed of Backbone (including ResNet and FPN), Region Proposal Network, ROI Align layer, Fully Connected Network, FC Layers, and other parts. The overall structure diagram is shown in Figure 1.

The cropped image is fed into the Backbone of the pre-trained network model, where ResNet101 causes the image size to decrease gradually and the feature map size to increase gradually. With the help of the FPN network structure, the feature layer is up-sampled, and multi-layer feature merging is carried out to obtain the more information-rich and effective feature layer, which is fed into the RPN to obtain the proposal box, which allows the network to conduct the initial selection of the proposal box.

(1)Targeted categorical regression task

The generated proposal box intercepts the effective feature layer to obtain the partial feature layer. Meanwhile, since the height and width of the proposal box are not fixed, the length and width of the partial feature layer are also not fixed, so it is also necessary to resize the partial feature layer to unify the operation of the subsequent network. All the above procedures take place in the ROI Align session. The partial feature layer is passed into the FC Layers (classification regression model) to determine whether the proposed box contains an object (classification task) and to fine-tune the proposal box to get a more accurate prediction box (regression task).

(2)Targeted semantic segmentation task

Since proposal boxes are usually large in number and not precise enough, the direct use of suggestion boxes for semantic segmentation tasks is ineffective, so prediction boxes are used to intercept the effective feature layer again and resize it by ROI Align to obtain fewer but more precise predicted boxes, which are finally fed into the Mask model in the FCN to achieve semantic segmentation of the target.

### 2.2. Backbone of Mask R-CNN

#### 2.2.1. ResNet Model

In the process of neural network model training, the core is to use CNN to extract the features information from the input image matrix; the deeper the network layers, the richer the abstract features that can be extracted to different levels, which contain more semantic features. However, the traditional convolutional layer or fully connected layer can never escape information loss and depletion during information transfer, and directly increasing the depth of the CNN can easily cause the phenomenon of gradient disappearance and gradient explosion, i.e., the model is degraded. To address the above problems, He [28] pioneered the ResNet.

The ResNet is modified from the VGG19 structure, with many branches in ResNet connecting inputs directly to a later layer to learn residuals. These simple equivalent mapping processes generate neither additional parameters nor added computational complexity. The above branch structure called shortcut that protects the information integrity, so that the whole network only needs to learn the difference part between the input and the output, which in turn effectively solves the existing gradient problem of the deep neural networks, simplifies the learning objective and difficulty, and enhances the network performance on the extraction and retention of features effectively.

ResNet contains two basic blocks, Identity Block and Convolution Block, where Identity Block is used when the dimensions of the inputs and outputs are the same, which can be connected in series and directly added, and is used to deepen the network, while Convolution Block is used when the dimensions of the inputs and outputs are not the same, which cannot be connected in series, and is used to change the dimensions of the network, and then connected in series to deepen the network. The schematic diagram of the two basic modules is shown in Figure 2.

#### 2.2.2. FPN Model

In deep neural networks, low-level features often contain more detailed information (color, contour, texture) with a lot of noise and irrelevant information. On the other hand, high-level features contain sufficient semantic information (categories, attributes, etc.) with low spatial resolution, so the loss of information on the high-level features is more severe. In the process of feature extraction, low-level and high-level features have different uses. If the advantages of both can be adequately combined, the effect of feature extraction can be significantly improved.

The Feature Pyramid Network (FPN) proposed by Lin et al. [29] constructed a series of images or feature maps at different scales for model training and evaluation, which aimed to improve the robustness of the detection algorithm for different sizes of detection targets. The structure of the FPN is shown in Figure 3. The FPN adopts a bottom-up and horizontally connected feature transfer method, which merges the lower-layer feature maps with less semantic information with the higher-layer feature maps with rich semantic information in order to significantly improve the feature utilization efficiency while maintaining a high training and prediction speed. However, there is a large computational overhead when the FPN calculation is directly based on the original definition. To reduce the computational overhead, FPN adopts a multi-scale feature fusion approach, which can substantially improve the scale robustness of the feature representation without substantially increasing the computational overhead.

### 2.3. ROI Align Model

The role of ROI pooling is to intercept the features of each ROI from the feature map, which is used to convert the ROIs’ features of different sizes to the same size to ensure that it can be connected to the lower fully connected layer. There is an evident defect in the original ROI pool layer, i.e., two quantization errors are generated in each pool rounding process: the first is from the image coordinate system to the feature map coordinate system, and the second is from the feature map coordinate system to the ROI coordinate system in the feature map. These will have a significant negative impact on the subsequent instance segmentation after the above errors are gradually accumulated and scaled up.

To alleviate the problem of excessive quantization error accumulation in ROI pooling, the ROI Align layer is used in the Mask-RCNN, replacing the traditional quantization rounding operation with bilinear interpolation, which makes full use of the four real-existing integer pixel values around the unrounded virtual pixel points in the original map to jointly determine a pixel value in the target map. Compared with the original ROI pooling, changing to the ROI Align layer can significantly reduce the calculation error and improve the overall accuracy of the model.

### 2.4. Loss Function

The loss function, as one of the critical parameters determining the predictive performance of deep neural networks, is used to assess the discrepancy degree between the predicted value and the true value of the model and to adjust the parameters of each node of the neural network through back-propagation to minimize this discrepancy, which largely determines the convergence effect of the model and also controls the objective of the network. The smaller the loss function value, the better the model performance.

The loss function of the Mask R-CNN mainly includes classification loss $L_{cls}$, bounding box regression loss $L_{box}$, and mask segmentation loss $L_{mask}$, which is represented by Equation (1).
(1)$$L=L_{cls}+L_{box}+L_{mask},$$

The cross-entropy loss function and Smooth L1 Loss are used in Mask R-CNN to compute the category loss and bounding box regression loss, respectively, where the Smooth L1 Loss is a combination of the L1 loss function and the L2 loss function, which was proposed by Girshick R [7] in the Fast R-CNN paper, to solve the problems of non-conductivity of the L1 loss function at 0, and the gradient of L2 loss function is easy to explode when the predicted value differs greatly from the real solved value, as shown in Equation (2).
(2)$$smooth_{L_{1}}(x)=\left\{\begin{array}{l} 0.5x^{2}\;\left|x\right|<1\\ \left|x\right|-0.5 \end{array}\right.,$$

## 3. Improvement of Mask R-CNN Network

With the addition of FPN and ROI Align, Mask-RCNN has a high instance segmentation accuracy for routine target detection tasks. However, Backbone in traditional Mask R-CNN often performs poorly when dealing with micro-crack detection tasks. It is easy to have inaccurate detection frames surround in the detection stage and inaccurate segmentations in the target segmentation stage. Therefore, this section will focus on the above problems to optimize and improve the Mask R-CNN structure to improve the model feature extraction performance.

### 3.1. Improved FPN Model

The Backbone of Mask R-CNN uses the ResNet50/101 + FPN scheme. Although the FPN module combines deep and shallow features, it still has the problem of insufficient utilization of multi-scale features.

The original feature map transfer roadmap of FPN is shown in Figure 4a, where blue arrows represent the top-down feature fusion path. When dealing with the detection task of small defect targets, it is difficult for only one feature fusion path to make sufficient use of the underlying detail features, and the long connection path from the bottom layer to the top layer weakens the underlying detail features, reducing the feature extraction performance. Therefore, based on the original feature map transfer roadmap of FPN, two additional feature fusion paths are added in this section, as shown in Figure 4b, where the orange arrow indicates a bottom-top feature fusion path, and the red arrow indicates an additional top-bottom feature fusion path.

In the original feature map transfer roadmap of FPN (Figure 5a), $P_{2}\sim P_{6}$ is used as an effective feature layer for the network to get the suggestion box, but in the improved feature map transfer roadmap in this section, the original $P_{2}\sim P_{6}$ will be used as an intermediate quantity and continue to be passed backward. We write $P_{2}\sim P_{6}$ as $F_{2}^{\prime }{\sim F}_{6}^{\prime }$. Firstly, adjust $F_{i}^{\prime }$ to the same size as $F_{i+1}^{\prime }$ by using the 3 × 3 convolution kernel with step size 2, directly add $F_{i}^{\prime }$ and $F_{i+1}^{\prime }$ pixel by pixel (the channel numbers of $F_{2}^{\prime }{\sim F}_{6}^{\prime }$ are all 256) to obtain the effective feature layers $X_{2}\sim X_{5}$ (where $X_{2}$ is directly obtained from $F_{2}^{\prime }$). Then, an effective feature layer with a higher resolution will be obtained through 2× up-sampling again to $X_{3}\sim X_{5}$. Then, when the size of $X_{i+1}$ is consistent with $X_{i}$, we add $X_{i}$ and $X_{i+1}$ together, and a 3 × 3 convolution operation is performed on the new fusion feature layer to reduce the aliasing effect caused by the up-sampling process. The final new effective feature layers $P_{2}\sim P_{6}$ is obtained ($P_{5}$ is obtained from $X_{5}$ directly through 3 × 3 convolution operation, and $P_{6}$ is obtained from $X_{5}$ through maximum pooling).

Compared with the original FPN effective feature layers, the new effective feature layers $P_{2}\sim P_{6}$ obtained through the improved feature map transfer roadmap have richer semantic and detailed information, which is conducive to improving the detection and location accuracy of surface micro-cracks.

### 3.2. Improved ResNet Model

For micro-defect detection tasks, it is usually necessary to extract and combine feature information and context information of different scales to identify target objects, so perceiving different scales’ information is crucial for target classification and semantic segmentation. The basic micro-crack’s context information often occupies a larger area than the micro-crack itself. For example, when the micro-crack exists on a slab, the deep learning network model will better judge whether the abnormal area is a micro-crack according to the context of the large-scale slab.

To enhance the capability of the feature extraction network at different scales, this section draws on the parallel network idea of GoogLeNet to construct hierarchical residual connections within a single residual block of the ResNet, so that a residual block can realize the feature extraction of multiple receptor fields. Figure 5a is the residual module in ResNet, and Figure 5b is the improved hierarchical residual module proposed in this section.

It can be seen from Figure 5b that this section replaces the original 3 × 3 convolution blocks with a set of smaller convolution blocks. When the input feature graph X is convolved by 1 × 1, the channel dimension changes and the input is divided into *s* groups (where *s* = 4), where each group of input is represented by $X_{i}$. The 3 × 3 convolution of group i is represented by $K_{i}$, and the output of group i after $K_{i}$ is represented by $Y_{i}$, where $X_{1}$ can directly obtain $Y_{1}$ without undergoing 3 × 3 convolution, and $X_{2}\sim X_{4}$ can obtain $Y_{2}\sim Y_{4}$ after undergoing 3 × 3 convolution. At the same time, the local residual structure is added in the process of obtaining $Y_{4}$ from $X_{3}$ and $X_{4}$ in order to increase the scales’ number that can be expressed by the output features. $X_{i-1}$ is fed into $K_{i}$ after adding the output of the 3 × 3 convolution to $X_{i}$. The above process can be shown as Equation (3).
(3)$$Y_{i}=\left\{\begin{array}{l} X_{i}\;i=1\\ K_{i}(X_{i})\;i=2\\ K_{i}(X_{i}+Y_{i-1})\;2<i\leq s \end{array}\right.,$$

Output Y can be obtained by splicing the outputs of *s* parts according to the channel dimension. The channel dimension is raised by 1 × 1 convolution layer, and then added with residual branches. Output F can be obtained with the ReLU activation function. By establishing hierarchical residual connections within residual blocks, the feature information of multiple scales can be obtained within one residual block, as shown in Equations (4)–(7), where ∗ represents the symbol of the convolution operation.
(4)$$Y_{1}=X_{1},$$
(5)$$Y_{2}=X*(3\times 3),$$
(6)$$Y_{3}=[X_{3}+X_{2}*(3\times 3)]*(3\times 3)=[X_{3}+Y_{2}]*(3\times 3),$$
(7)$$Y_{4}=\{X_{4}+[X_{3}+X_{2}*(3\times 3)]*(3\times 3)\}*(3\times 3)=[X_{4}+Y_{3}]*(3\times 3),$$

Each $K_{i}$ can receive the output features of the previous convolution layer, resulting in a larger receiving domain than the original 3 × 3 convolution operation, and the output of the hierarchical residual structure can contain more scale context information due to the combined explosion effect.

### 3.3. Deformable Convolution Model

In mathematics, the standard definition of a convolution kernel is the integral of the two functions’ product after inversion and shift, as shown in Equation (8). Usually, the function g is referred to as the filter, and the function f is referred to the raw data of the signal or image.
(8)$$(f*g)(t)=\int_{-\infty }^{+\infty }f\left(\tau \right)g\left(t-\tau \right)d\tau ,$$

In CNN, the convolution kernel is essentially a filter. When processing images, the convolutional kernel is used to obtain a weighted average of pixel values in a small region in a given input image, and the output image result is the corresponding pixel value. It can extract different local features and generate lots of neurons by setting up different forms of convolutional kernels. It can construct a convolutional neural network through deep connection.

In general, a larger convolution kernel obtains a larger receptive field, meaning more picture information can be seen and more global features can be obtained. However, a large convolution kernel will lead to a significant increase in computation and a decrease in performance. The traditional convolution kernel size is usually a standard rectangle, such as 1 × 1, 3 × 3, 5 × 5, etc. In the actual target detection task, the convolution kernel size can be adjusted according to the target characteristics. However, when the target shape changes, or the shape is irregular (for example, when the length and width of the target are very different, such as micro-cracks), the traditional rectangular convolution kernel easily misses the target features during the convolution process, which reduces the feature extraction performance. Therefore, a deformable convolution kernel is added in this section to replace the rectangular convolution kernel in the original Mask R-CNN to improve the feature extraction performance for small targets.

The idea of deformable convolutional networks was proposed by Dai et al. [30] in an ICCV 2017 article, and in this article the authors proposed a deformable convolutional network called DCNv1. The core idea is to add an offset direction parameter based on the traditional convolution kernel and learn the convolution kernel and offset simultaneously. The calculation process from standard convolution kernel to deformable convolution kernel can be shown as Equations (9) and (10), where R is a set of pixels sampled from the input feature map, $P_{0}$ is the center point in the feature map, $P_{n}$ is a point in the feature map, w is the weight of convolution kernel, and $\Delta P_{n}$ is the shift matrix of point $P_{n}$. The schematic diagram is shown in Figure 6. Deformable convolutional kernel DCNv1 can bring a freer receptive field, easily replace standard modules in existing CNNs, and learn the effectiveness of dense space transformation for complex visual tasks in deep convolutional neural networks.
(9)$$y(P_{0})=\sum_{P_{n}\in R}w(P_{n})\cdot x(P_{0}+P_{n}),$$
(10)$$y(P_{0})=\sum_{P_{n}\in R}w(P_{n})\cdot x(P_{0}+P_{n}+\Delta P_{n}),$$

However, the offset module in DCNv1 generate a large amount of context-free information, which is unfavorable for small target detection tasks. Wang et al. [31] proposed DCNv2, which adds more deforming convolution layers and allows the model to learn not only the offset but also the weight of each sampling point, effectively reducing the interference of irrelevant information. The calculation process of DCNv2 from input to output can be expressed by Equation (11), where $\Delta m_{n}$ in Equation (11) is the weight coefficient added on the basis of Equation (10), and its value range is [0,1].
(11)$$y(P_{0})=\sum_{P_{n}\in R}w(P_{n})\cdot x(P_{0}+P_{n}+\Delta P_{n})\cdot \Delta m_{n},$$

Based on the modification of the residual structure of ResNet in Section 3.1, the standard 3 × 3 convolution in the hierarchical residual structure is replaced with deformable convolution, as shown in Figure 7a,b.

In DCNv2, all 3 × 3 convolution kernels are replaced with deformable convolution from conv_3 to conv_5 of the ResNet, which achieves good performance improvement in the COCO dataset. However, with the increase in network depth, the irregularity of target features gradually decreases in the process of layer-by-layer convolution. Therefore, considering the computation increase and the performance improvement after adding convolutional layers, this section only replaces the 3 × 3 convolution with deformable convolution in conv_2 and conv_3 of the ResNet in Mask R-CNN, keeping conv_1, conv_4, and conv_5 unchanged. The improved ResNet structure is shown in Figure 7c, where Res-DCN represents a hierarchical residual module with deformable convolution added.

### 3.4. Attention Mechanisms Model

In the micro-defect target detection task, micro-cracks account for a small proportion of pixels in the image, and are usually elongated in shape, with weak visual features, and easily interfered with by complex backgrounds. The effective feature information that can be extracted from images in deep convolutional networks is very limited and is more likely to be lost. Therefore, this section adds an attention mechanism based on the original Mask R-CNN to enhance the network’s target attention to small defects and to appropriately weaken its attention to other information.

The addition of attention mechanisms in neural networks can flexibly adjust the proportion of weight value to allocate computing resources to more critical tasks and reduce the attention paid to other information in the case of limited computing power in order to improve the efficiency and accuracy of task processing.

Mainstream attention mechanism models mainly include channel attention, spatial attention, CBAM attention mechanism, etc. Among them, the channel attention mechanism allocates different attention weights according to the difference in the importance degree of other channels, while the spatial attention mechanism allocates different attention weights according to the difference in the importance degree of different regions. The purpose of both of them is to realize the rational allocation of model computing resources and to achieve as many effects as possible with limited resources.

CBAM module is a hybrid attention mechanism based on SENet, which combines the advantages of the channel attention mechanism and spatial attention mechanism. It can serialize the attention feature information in both channel and space dimensions, multiply the two features’ information with the original input feature, and generate the final feature after adaptive feature correction. CBAM is a lightweight module that can be embedded into any backbone network to improve performance, especially in cases where the features of different parts are particularly different. It is designed to enhance the ability of convolutional neural networks to focus on images.

In this section, two attention module embedding schemes are set up to explore the effect of attention mechanism on improving network performance. The schematic diagram of the two schemes is shown in Figure 8. For scheme (a), the CBAM module is arranged after the first Max Pooling layer of the ResNet structure. For scheme (b), the CBAM module is embedded between the third and the fourth convolutional block of the ResNet structure. The above two different embedding schemes are intended to compare the influence of embedding attention mechanism modules in different locations of ResNet on feature extraction ability.

### 3.5. Improved Loss Function

Smooth L1 Loss, the traditional Mask R-CNN loss function, assumes that the four points of the boundary box are independent of each other when calculating the boundary box regression loss of target detection and calculates the loss values of the four points, adding them together to obtain the final boundary box regression loss. However, these four points are interrelated, and the indicators used in the actual evaluation are also calculated based on the intersection ratio (IOU) between the predicted border and the actual border, which does not match the calculation method of Smooth L1 Loss; in addition, there are often situations where the loss values of multiple detection boxes are close, but the IOU values are very different. Therefore, in this section, we analyze whether using IOU to design the Loss function is more in line with the idea of boundary box regression, and intent to replace the loss function Smooth L1 Loss in the original Mask R-CNN with the IOU Loss Function.

Common IOU Loss Functions include IOU, GIOU, DIOU, and CIOU, as shown in Equations (12)–(15):(12)$$L{}_{IOU}=1-\frac{C}{A+B-C},$$
(13)$$L{}_{GIOU}=1-(\frac{C}{A+B-C}-\frac{D}{E}),$$
(14)$$L{}_{DIOU}=1-(\frac{C}{A+B-C}-\frac{d}{L}),$$
(15)$$L{}_{CIOU}=1-(\frac{C}{A+B-C}-\frac{d}{L}-\alpha \nu ),$$
where *A* is the area of the real rectangular box, *B* is the area of the predicted rectangular box, *C* is the area of the intersection area of *A* and *B*, and *D* is the area that does not intersect within the range of the external rectangle of *A* and *B*. *E* is the area of the external rectangle of *A* and *B*, *d* is the Euclidean distance between the center points of *A* and *B*, *L* is the diagonal distance of the external rectangle *E*, *α* is the weight coefficient, and *ν* is the similarity index of the aspect ratio between the predicted box and the real box.

Compared with Smooth L1 Loss, IOU Loss establishes box-based loss calculation, which can directly reflect the comparison effect between the predicted box and the real box. However, when the predicted box and the real box have no intersection area, that is, when *C* = 0, the Loss function is not distinguishable. Moreover, when the IOU Loss is the same, predicted and real boxes can intersect in various ways. Hence, the IOU Loss neither optimizes the absence of intersecting regions nor reflects the way prediction and real boxes intersect.

GIOU Loss adds the item $\frac{D}{E}$ based on IOU Loss. Considering the influence of disjointed areas in the external rectangular range, it can also carry out learning training even when there is no overlap area between *A* and *B*. GIOU has several advantages: (a) GIOU is scale invariant; (b) GIOU is the lower bound of the IOU, where the two boxes overlap indefinitely, IOU = GIOU = 1; (c) GIOU pays attention not only to overlapping areas, but also to other non-overlapping areas, which can better reflect the degree of overlap between two boxes.

DIOU improved the problem of GIOU. As shown in Figure 9a, the ratio $\frac{d}{L}$ of Euclidean distance d from the center point of the prediction box and the real box to the diagonal length of the external rectangle is introduced as a penalty term. If the IOU value is the same, this ratio can reflect the intersection between the prediction box and the real box. When the two boxes do not intersect, as shown in Figure 9b, the IOU value is 0, and the penalty item $\frac{d}{L}$ is the main optimization object, making the prediction box move to the real box until it coincides. When the two boxes are inclusive, as shown in Figure 9c, the penalty item $\frac{d}{L}$ is still the optimization object, guiding the prediction box to move in the direction of the coincidence of the center point with the real box.

The constraint term αν on-aspect ratio is added to CIOU Loss. Compared with DIOU Loss, the convergence speed is somewhat accelerated. However, involving the calculation of inverse trigonometric functions, the optimization speed of the model will be reduced to a certain extent. The author used the Faster RCNN model to conduct experiments on the MS COCO 2017 dataset, and the results showed that the accuracy of CIOU Loss was slightly lower than that of DIOU Loss for detecting small-scale targets.

Therefore, after comprehensively weighing the optimization speed and performance in small target detection tasks, this section uses DIOU Loss to replace the boundary box regression Loss Smooth L1 Loss in Mask R-CNN. Two modules in Mask R-CNN need to calculate boundary box regression Loss $L_{box}$, namely the RPN network and ROI Head module. The boundary box regression loss of both modules is calculated through Smooth L1 Loss. This section uses DIOU Loss to replace both Smooth L1 losses.

## 4. New Network Performance Experiment

### 4.1. Experiment Setup, Data Sources and Datasets

In order to verify the effectiveness and feasibility of the Mask R-CNN improvement scheme proposed above, this part plans to carry out data collection, data enhancement, and comparison and verification of model detection performance in sequence.

Generally, high-quality datasets are an important prerequisite for the generalization ability and robustness of deep learning models. The paucity of micro-crack datasets on the surfaces of metal structures largely restricts the improvement of the performance of deep learning models for micro-crack detection tasks. Therefore, in this section, we self-construct a micro-crack dataset on the surfaces of metal structures based on existing laboratory conditions.

Images are stored as grayscale values, with each pixel corresponding to a set of grayscale values. The more pixels a defect feature occupies in an image, the greater the corresponding set of gray values, which means that the defect feature contains more information. In order to contain as many defect features as possible in each micro-crack image and to enable the model to extract the defect features effectively, we shot the micro-crack images at a higher resolution in the image acquisition stage. In this section, the selected camera was a CANON EOS 70D digital SLR camera, the optical lens was a CANON EF-S 18–200 mm f/3.5–5.6 IS, and the resolution of the collected crack images was uniformly set to 1920 × 1280. To improve the diversity of the dataset, micro-cracks on the surface of test CT specimens, large and complex mechanical structures, and other objects were collected. All original images in the dataset were exported directly after being taken with the digital SLR camera. Finally, the initial dataset of small crack images was collected and obtained, containing 500 images. In the micro-crack dataset, we specially designed some images with a lot of complex disturbing objects in the background so as to improve the detection performance of the model in complex-background environments.

Since the small size of the acquired image data is insufficient to train a deep learning model with good performance, data enhancement of the original micro-crack image dataset is needed to improve data diversity and dataset size. In this section, non-geometric transformation operations such as adding Gaussian noise, adding pretzel noise, randomly setting the black point, randomly deleting channels, changing the color space, changing the color temperature, linear contrast, and geometric transformation operations such as rotating, translating, randomly splicing, and mirroring were performed on the original image obtained from the acquisition to avoid the loss of micro-crack feature information caused by direct cropping. A partial display of the original images and the enhanced images in the micro-crack dataset is shown in Figure 10.

The image number in the micro-crack dataset was increased from 500 to 2000 through the above data enhancement method. The current common small dataset divides the training, validation, and test sets with a ratio of 6:2:2. To improve the training effect of the model, this section appropriately reduces the test set and suitably increases the training set, dividing the training, validation, and test set ratio of the micro-crack dataset into a ratio of 7:2:1.

Then, based on Labelme in the Python environment, the training and validation set images in the dataset are manually labeled with micro-crack features and converted to COCO format, and the labeling effect is shown in Figure 11. To better display the annotation effect, some images are enlarged.

### 4.2. Experimental Environment and Evaluating Indicator

#### 4.2.1. Experimental Environment

To ensure that the network models used and optimized can be trained efficiently, this section builds the necessary experimental environment, details of which are shown in Table 1.

#### 4.2.2. Evaluating Indicator

To quantitatively evaluate the overall performance of a target detection model for a particular detection task, a set of evaluation metrics must be specified in advance, typically including Accuracy, Precision, Recall, and Average Precision, etc., where Accuracy is the proportion of the number of total samples in which the prediction result is positive, Precision is the percentage of the number of total samples predicted as positive that are actually ground truth samples, and Recall is the percentage of the number of total ground truth samples that are predicted as positive samples, as shown in Equations (16)–(18). Table 2 shows the confusion matrix representation of TP, FP, FN, and TN, where TP is the statistic value for positive samples correctly identified as positive samples, FP is for negative samples incorrectly identified as positive samples, FN is for positive samples incorrectly identified as negative samples, and TN is for negative samples correctly identified as negative samples.
(16)$$Accuracy=\frac{TP+TN}{TP+TN+FP+FN},$$
(17)$$Precision=\frac{TP}{TP+FP},$$
(18)$$Recall=\frac{TP}{TP+FN},$$

Moving the IOU threshold stepwise from 0 to 1 produces a set of Precision and Recall values from which a P–R curve can be plotted to show the relationship between P and R. The area under the P–R curve is the AP value, which is positively correlated with the performance of the model, that is, the larger the area enclosed by the P–R curve, the better the performance of the model is generally considered. At the same time, we also want the Precision value corresponding to any Recall value in the model P–R curve to be as high as possible. When evaluating models, it is usually assumed that the higher the AP value of a category, the better the model recognizes objects in that category. The average of the AP values of all categories, i.e., the mAP value, is expressed in Equations (19) and (20); the higher the mAP value, the better the overall results achieved by the model in all categories.
(19)$$AP=\int_{0}^{1}P\left(R\right)dR,$$
(20)$$mAP=\frac{1}{N}\sum_{i=0}^{N}{AP}_{i},$$

#### 4.2.3. Network Training Parameters

Before training a deep learning model, appropriate training hyperparameters must be specified to ensure that the neural network model neither fails to fit nor overfits during the training phase, while allowing the network to learn the data structure features as quickly as possible. Epochs, learning rate, and batch size are the most important hyperparameters for training a deep learning model. According to the network model used in this paper and the image characteristics of the datasets, specific hyperparameters are set as shown in Table 3.

### 4.3. Experimental Results and Analysis

#### 4.3.1. Scheme Design of Network Performance Comparison Experiment

To verify the effect on the network performance of each optimization scheme proposed above, this section sets up a comparative ablation experiment. To facilitate the distinction, the model that incorporates all the improvement schemes is denoted as Mask-All; the model that deletes the FPN structural improvement scheme is denoted as FPN-Off. In this paper, two attention mechanism improvement schemes are proposed. The first one is to embed the CBAM module at the two ends of the ResNet, and the model that deletes this scheme is denoted as Att-1-Off; the second one is to embed the CBAM module at the middle of ResNet, and the model that deletes this scheme is denoted as Att-2-Off. The model that deletes the deformable convolution module is denoted as DCNv2-Off; the model that deletes the loss function improvement is denoted as DIOU-Off; the model that deletes the residual module improvement is denoted as Res-Off; and the original Mask R-CNN model is used as a control group for the ablation experiment.

#### 4.3.2. Comparative Analysis of Network Detection Performance

Firstly, the missed and false detection performance evaluation of each model is carried out, where the missed detection number refers to the number of images that are not detected as containing a target but actually contain targets, and the false detection number refers to the number of images that do not contain a target but are detected as containing targets. We randomly selected 150 images from the micro-crack dataset as a test set, counting the number of correct detections, missed detections, and false detections of each model; the results are shown in Figure 12.

It can be seen from Figure 12 that the number of correct detections of Mask-All is 118, which is a more obvious performance improvement compared to the original Mask R-CNN model, thus proving that mixing all the optimization and enhancement schemes proposed in this paper can improve the detection performance of the original Mask R-CNN model.

DIOU-Off, DCNv2-Off, and Res-Off all show a decrease in the correct detections’ number compared to Mask-All, indicating that the optimization schemes for the loss function, deformable convolution, and hierarchical residual network structure all have some improvement effect on the detection performance of the model. The false detections’ number for FPN-Off is higher by 16 compared to Mask-All, indicating that the optimization schemes for the FPN structure can have a more significant impact on the overall detection performance of the model than the other schemes.

The improvement scheme for the attention mechanism produced some interesting phenomena: the Att-1-Off model slightly increased the number of correct detections compared to Mask-All to 119, i.e., the improvement scheme appeared to have a counterproductive effect on the model’s detection performance. Attempts to analyze the cause of this phenomenon at the structural level revealed that at the head position of ResNet, the larger feature layer size and a smaller channels number lead to a larger range of spatial attention mechanisms that make it difficult to accurately target the location of the target, and the overall generalization of the channel attention mechanisms is lower. At the tail part of ResNet, the larger number of channels leads to a higher risk of overfitting, and the smaller feature layer sizes result in the convolution operation introducing more non-pixel information. Therefore, it is not easy to achieve the desired optimization effect by placing the CBAM module at the head and tail ends of ResNet. But on the other hand, the small size of the test set used in this paper may also be one of the main reasons for the above phenomenon.

For Att-2-Off, the correct detections’ number decreased by 4 compared to Mask-All, and the missed detections’ number increased to 24, which is significantly higher; this shows that embedding the CBAM module in the middle of the ResNet can positively affect the model’s detection performance.

The results of the above analysis show that, apart from whether the placement of the CBAM module at the head and tail of the ResNet could promote the model detection performance, all the optimizing schemes proposed in this paper can improve the detection performance of the model.

#### 4.3.3. Comparative Analysis of Convergence Performance in Network Training

The Loss convergence during the model training process is one of the indicators used to evaluate the loss function and the overall performance of the model. To more clearly assess the extent of the impact of each improvement scheme on the model training process, the loss function curves change with the number of epochs in the training of the four models, namely the original Mask R-CNN, DIOU-Off, FPN-Off, and Mask-All, are plotted as shown in Figure 13.

It can be concluded that the loss curves of the original Mask R-CNN model show a rapid decline in the first part of the training process, but there is a sustained and more pronounced oscillation for about 10–90 epochs, with an overall convergence from the 90th epoch onwards.

Compared with the original Mask R-CNN, Mask-All has a better convergence effect of the loss function. Although the convergence speed slows down in about 15–20 epochs, the overall curve is gentle and fast convergence; the overall convergence tends to converge from the 25th epoch onwards, which proves that Mask-All, which integrates all the optimization schemes, has a more obvious improvement in the training process compared with the original Mask R-CNN.

The performance of the FPN-Off curve in the first training part is not quite consistent with other curves. For about the first 15 epochs of the loss function, convergence is relatively slow, and there is a certain degree of oscillation, which then quickly reaches the convergence and stabilization; the overall convergence curve is relatively smooth, even earlier than the Mask-All curve convergence. The above phenomena show that the FPN structure optimizing has no major effect on the model’s detection performance.

The DIOU-Off curve is consistent with that of the original Mask R-CNN, and there is a continuous and obvious oscillation for about 10–90 epochs, which indicates that the optimization scheme of the DIOU loss function has a significant effect on the model training speed. At the same time, observing the loss curves of Mask-All and FPN-Off also proves that the use of the DIOU Loss instead of Smooth L1 Loss has a specific accelerating effect on the convergence of the model’s loss function during training.

The above analysis shows that, among all the optimization schemes proposed in this paper, the optimization scheme for loss function can produce the most obvious improvement effect on model detection performance, and also proves the effectiveness of the optimization and improvement scheme proposed in this paper.

#### 4.3.4. Comparative Analysis of Network Object Recognition Rate

Further bar graphs of mAP values for all models are plotted to evaluate the impact of each optimization scheme on model recognition accuracy, as shown in Figure 14. It can be concluded that, compared with the original Mask R-CNN, the Mask-All mixed with all the improved schemes has the highest mAP value, reaching 78.6%.

The mAP value of Att-1-Off is 78.4%, which is not a significant decrease from Mask-All, indicating that embedding the CBAM module at the head and tail parts of the ResNet does not significantly improve the model’s detection accuracy.

The mAP value of Att-2-Off decreased to 76.2%, which is 2.4% less than that of Mask-All, indicating that embedding the CBAM module in the middle part of the ResNet can achieve a more desirable performance improvement effect, which is also basically consistent with the performance of Att-1-Off on the test set.

The mAP value of FPN-Off decreased to 74.2%, which is the lowest value of all the ablation single improvements, and decreased by 4.4% compared to Mask-All, indicating that the addition of bottom-up branches in the FPN structure can more significantly improve the mAP value of the model on the surface micro-cracks dataset.

The mAP values of DCNv2-Off, DIOU-Off, and Res-Off decreased by 1.5%, 2.3%, and 3.7%, respectively, compared to Mask-All, indicating that each of the remaining improvements has a positive effect on enhancing the detection performance of the model.

The above analysis shows that, among all the optimization schemes proposed, the improvement scheme for the FPN structure can produce the most obvious effect on the model detection accuracy, and also proves the effectiveness of the optimization scheme proposed in this paper.

#### 4.3.5. Comparative Analysis of Network Object Location Performance

In addition to being able to accurately identify a target, a good model should be able to simultaneously localize it accurately. Therefore, this section compares the target localization performance of the above models.

Ten small crack images that were correctly detected by all models simultaneously were randomly selected, and the real coordinate information of the micro-crack target that had been labeled using Labelme in each image was firstly read and collated into Table 4 ($x_{1}$, $y_{1}$, $x_{2}$, $y_{2}$ are the horizontal and vertical coordinates of the upper-left and lower-right points of the labeled rectangular box, respectively). As the dataset of each model is in COCO format, the test file gives the coordinates of the prediction box in the form of (*x*, *y*, *w*, *h*), i.e., the horizontal and vertical coordinates of the point at the upper-left corner of the prediction box and the width and height of the prediction box, which need to be converted according to Equations (21) and (22) to obtain the coordinates of the point at the lower-right corner.
(21)$$x_{2}=x_{1}+l$$
(22)$$y_{2}=y_{1}+w$$

The coordinate position data of the original Mask R-CNN and the hybrid improved Mask-All for the prediction box of the micro-cracks target in 10 images are read, respectively. Table 5 and Table 6 are summarized, combined with the real position coordinates of the rectangle boxes in Table 4; the localization error of the Mask R-CNN before and after the optimization can be obtained for the micro cracks target in the test images as shown in Equations (23)–(28):(23)$$\Delta x_{1}=\left|x_{1}^{\prime }-x_{1}\right|$$
(24)$$\Delta x_{2}=\left|x_{2}^{\prime }-x_{2}\right|$$
(25)$$\Delta y_{1}=\left|y_{1}^{\prime }-y_{1}\right|$$
(26)$$\Delta y_{2}=\left|y_{2}^{\prime }-y_{2}\right|$$
(27)$$\Delta \bar{x}=\frac{1}{2}\left(\Delta x_{1}+\Delta x_{2}\right)$$
(28)$$\Delta \bar{y}=\frac{1}{2}\left(\Delta y_{1}+\Delta y_{2}\right)$$
where $∆x_{1}$ denotes the absolute value of the error between the horizontal coordinates of the point in the upper left corner of the prediction frame and the true coordinates, and similarly, ${∆y}_{1}$, $∆x_{2}$, and $∆y_{2}$ denote the absolute value of the error between the vertical coordinates of the upper left corner of the prediction frame, the horizontal coordinates of the lower right corner, and the vertical coordinates of the lower right corner of the prediction frame and the true coordinates, respectively. To obtain the overall error of all the prediction frames, we take the average of $∆x_{1}$ and $∆x_{2}$, denoted as $∆\bar{x}$, and the average of ${∆y}_{1}$ and $∆y_{2}$, denoted as $∆\bar{y}$. However, there is no absolute correlation between $∆\bar{x}$ and $∆\bar{y}$. The positioning error of the detection frame is also related to the absolute size of the crack, i.e., if $∆\bar{x}$ is the same, the longer the length of the real frame of the crack, the smaller the error; if $∆\bar{y}$ is the same, the wider the width of the real frame of the crack, the smaller the error. Therefore, the ratio of $∆\bar{x}$ and the true frame length *l* and the ratio of $∆\bar{y}$ and the true frame width *w* are further derived, and the average of these two ratios is taken as the integrated localization error, which is denoted as ∆xy, as in Equation (29). Finally, the mean value of ∆*xy* of all the prediction frames in 10 images of the original Mask R-CNN and the hybrid improved Mask-All is taken as the overall error, which is shown in the last column of data in Table 5 and Table 6, and is used as a measure of the overall level of the model’s ability to localize towards the small cracks in the range of 10 images before and after the improvement.
(29)$$\Delta xy=\frac{1}{2}\left(\frac{\Delta \bar{x}}{l}+\frac{\Delta \bar{y}}{w}\right)\times 100\%$$

From the experimental results of the comparison between the original Mask R-CNN and the improved Mask R-CNN for micro-crack localization performance, it can be seen that in the 10 randomly selected micro-crack images, the overall localization error of Mask R-CNN for micro-cracks reaches 17.22%. In contrast, the localization error of the fused multi-improved Mask-All is only 7.44%, which is 9.78% lower than that of the original Mask R-CNN. It proves that the fusion of several improvements has improved the localization accuracy of Mask R-CNN for micro-cracks.

In the process of verifying the model positioning performance by using 10 randomly selected diagrams (as shown in Table 4, Table 5 and Table 6), we found that compared with the real crack positioning, the original and the improved Mask R-CNN have the largest positioning errors in Image No. 1, Image No. 621, Image No. 992, and Image No. 1065. The crack regions of the above four images are enlarged and shown in Figure 15.

In Image No. 1, there are surface light mutations around the crack, which increases the difficulty of identification. In Image No. 621, the crack features have become blurred under the multiple effects of changing the color space and adding noise. Image No. 992 has low overall clarity, small crack size, and less characteristic information. In Image No. 1065, the gray value of the crack is similar to that of the surrounding background, which makes it relatively difficult to distinguish. As can be seen from Figure 15, compared with the original Mask R-CNN, the improved Mask R-CNN can better avoid the background region in the process of crack identification and more accurately frame the crack body, thus proving that the improved scheme proposed in this paper is feasible.

## 5. Conclusions

Micro-cracks on the surfaces of metal structures have weak target features and are easily disturbed by complex backgrounds. When using deep convolutional neural network to detect micro-cracks, they are easily missed and falsely detected. Therefore, this paper first makes a surface crack dataset, including metal test samples and metal structural parts of large equipment, and enhances the dataset by adding noise, geometric transformation, and other operations. Then, based on the original Mask R-CNN, this paper explores an improved method for micro-crack detection tasks on the surface of metal structural parts.

In this paper, the network structure of the Mask R-CNN is improved: based on the original FPN structure, a bottom-up feature fusion path is added to improve the information utilization rate of the underlying feature layer. The original ResNet was changed to a hierarchical residual structure to improve the efficiency of extracting features at different scales. A deformable convolution kernel replaces the 3 × 3 convolution kernel in the original residual structure to improve the feature extraction efficiency of the target with small and slender cracks. The CBAM attention module is tentatively embedded in the head, tail, and middle parts of ResNet to increase the expression weight of the micro-crack region in the feature layer. The original Smooth L1 Loss function was replaced with DIOU Loss to optimize the network training effect. Finally, an ablation experiment was conducted to verify the effect of each improvement scheme on the performance improvement of the model.

The performance comparison results of various optimization models show that all the improvement schemes proposed in this paper improved the performance of the original Mask R-CNN, among which the improvement of FPN had the most significant effect on the recognition rate and accuracy of network detection. Replacing the original Smooth L1 Loss with DIOU Loss can significantly improve the convergence effect during network training. The integration of all the improvement schemes can produce the most significant performance improvement effect in the aspects of identification, classification, and positioning at the same time, which proves the rationality and feasibility of the improved scheme in this paper.

On the other hand, the research process of this paper still has some limitations that cannot be ignored. In order to make each image in the micro-crack dataset contain as many crack details as possible, so that the model could extract the features of micro-cracks more efficiently, this paper used a digital SLR camera to capture micro-crack images with higher resolution in the image acquisition stage. However, the issue of how to effectively extract features based on low-resolution micro-crack images and train the detection model to achieve similar performance to the model proposed in this paper (using high-resolution micro-crack images for training) has not been explored.

Furthermore, in order to verify the detection performance of the improved Mask R-CNN trained in this paper for relatively low-resolution micro-crack images, the resolution of the dataset used for training was uniformly reduced from 1920 × 1280 to 640 × 480, and it was imported into the same model for prediction and inference. As the defect feature information contained in the micro-crack target in the image after resolution reduction is reduced synchronously, more micro-crack images appear to be false detection and missing detection (compared with the detection results before image resolution reduction). This comparison result also indicates that the improved method proposed in this paper still has a certain optimization space in terms of the defect feature extraction performance of the model.

At the same time, although the 500 original micro-crack images captured in this paper were enhanced to 2000 images via our data enhancement method, it is still difficult to support and train a micro-crack detection model with excellent performance. Therefore, it is very valuable to explore more efficient methods of enhancing small-scale datasets or model training based on small-scale datasets.

## Figures and Tables

**Figure 1 sensors-24-00062-f001:**
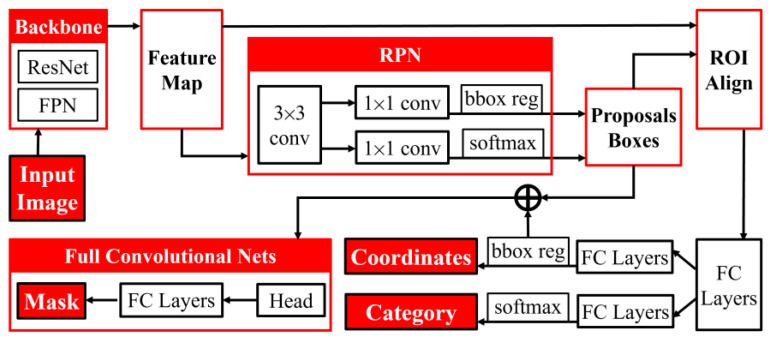
The overall structure diagram of Mask R-CNN.

**Figure 2 sensors-24-00062-f002:**
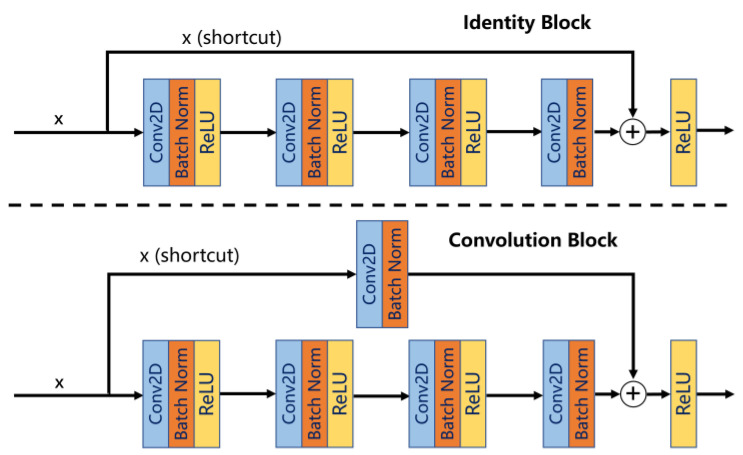
The schematic diagram of Identity Block and Convolution Block.

**Figure 3 sensors-24-00062-f003:**
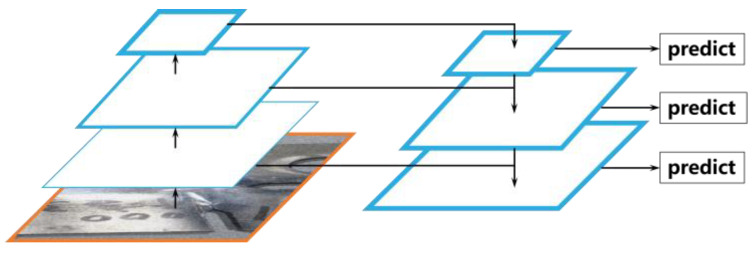
The structure of the Feature Pyramid Network.

**Figure 4 sensors-24-00062-f004:**
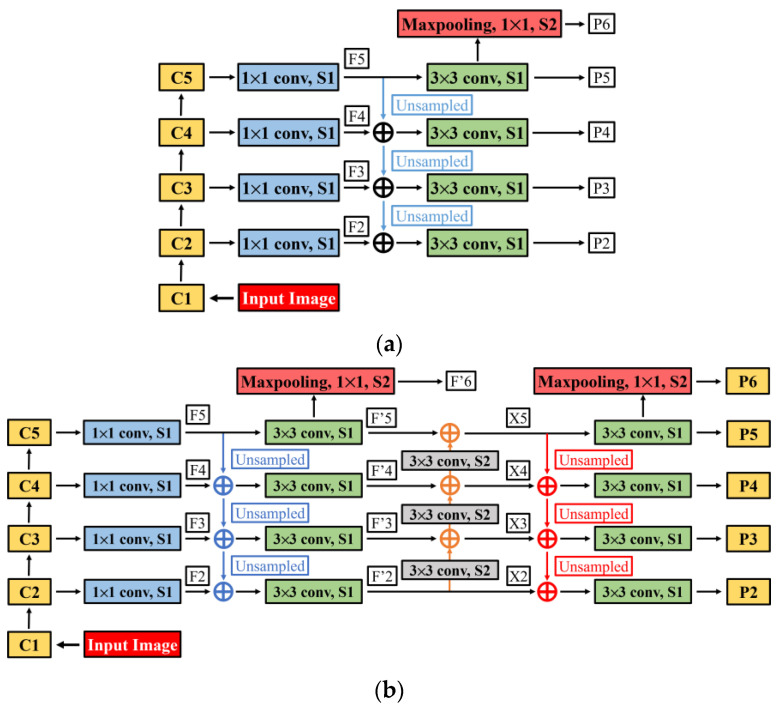
Feature map transfer roadmap of FPN. (**a**) Original feature map transfer roadmap of FPN. (**b**) Improved feature map transfer roadmap of FPN.

**Figure 5 sensors-24-00062-f005:**
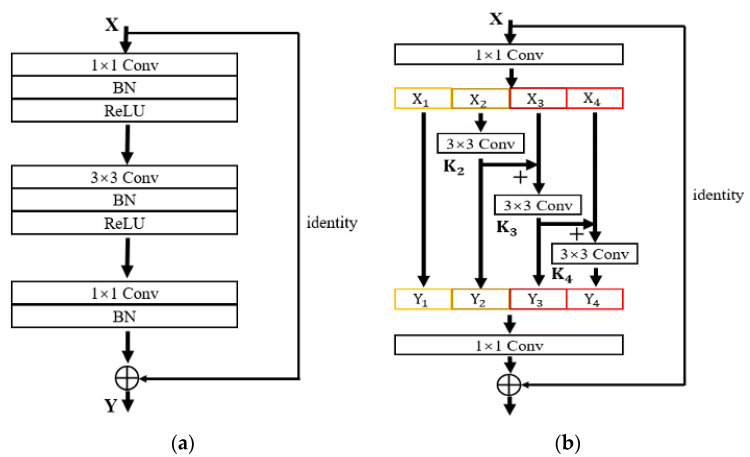
Residual module structure. (**a**) ResNet model. (**b**) Hierarchical residual module.

**Figure 6 sensors-24-00062-f006:**
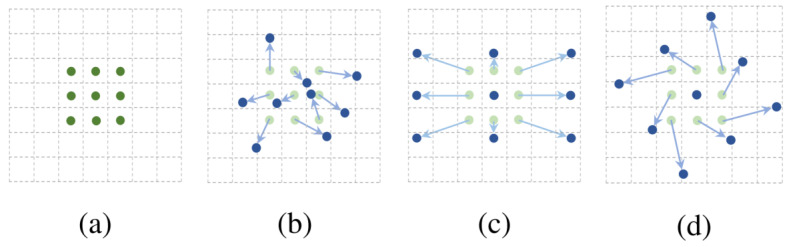
Illustration of the sampling locations in 3 × 3 standard and deformable convolutions. (**a**) Regular sampling grid (green points) of standard convolution. (**b**) Deformed sampling locations (dark blue points) with augmented offsets (light blue arrows) in deformable convolution. (**c**,**d**) Special cases of (**b**), showing that the deformable convolution generalizes various transformations for scale, (anisotropic) aspect ratio, and rotation.

**Figure 7 sensors-24-00062-f007:**
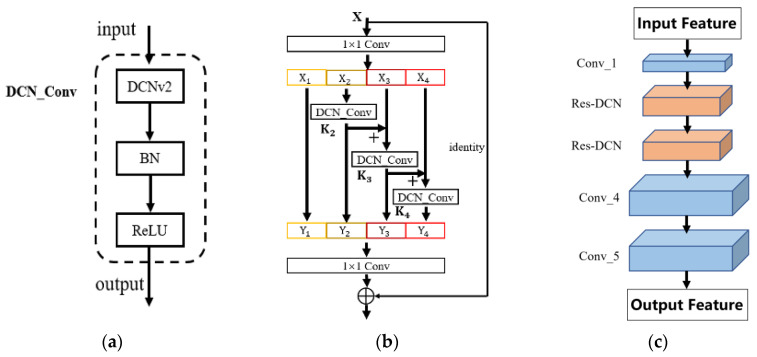
DCN_Conv kernel and improved ResNet structure. (**a**) DCN_Conv kernel structure. (**b**) DCN hierarchical residual. (**c**) Improved ResNet structure.

**Figure 8 sensors-24-00062-f008:**
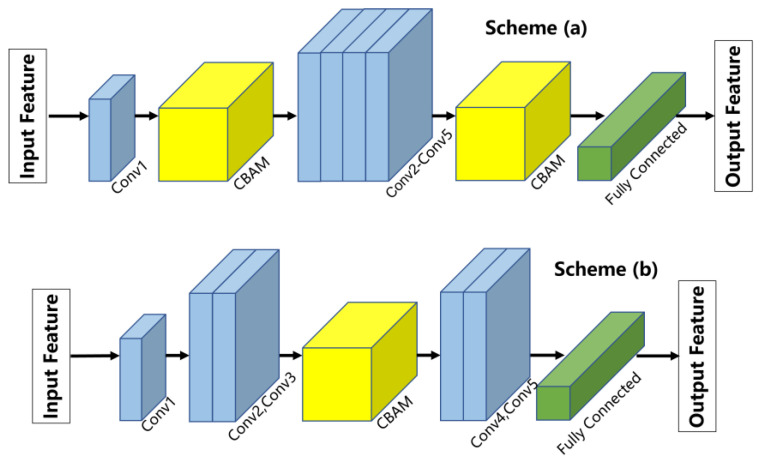
The schematic diagram of two different embedding schemes. Scheme (**a**) arranged the CBAM module after the first Max Pooling layer of the ResNet structure. Scheme (**b**) arranged the CBAM module between the third and the fourth convolutional block of the ResNet structure.

**Figure 9 sensors-24-00062-f009:**
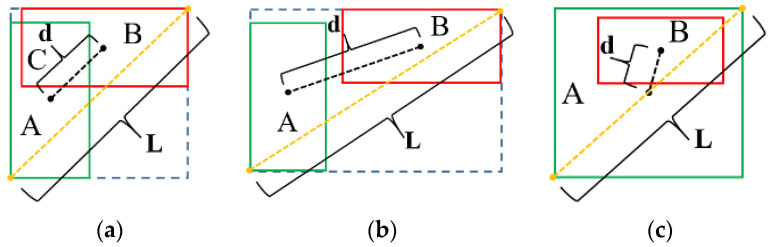
Diagram of DIOU Loss function. (**a**) Intersect. (**b**) No intersect. (**c**) Contain. The real rectangular boxes (marked as *A*) are shown in green, the predicted rectangular boxes (marked as *B*) are shown in red, the Euclidean distance between the center points of *A* and *B* (marked as *d*) are shown in black, the diagonal distance of the external rectangle (marked as *L*) are shown in orange.

**Figure 10 sensors-24-00062-f010:**
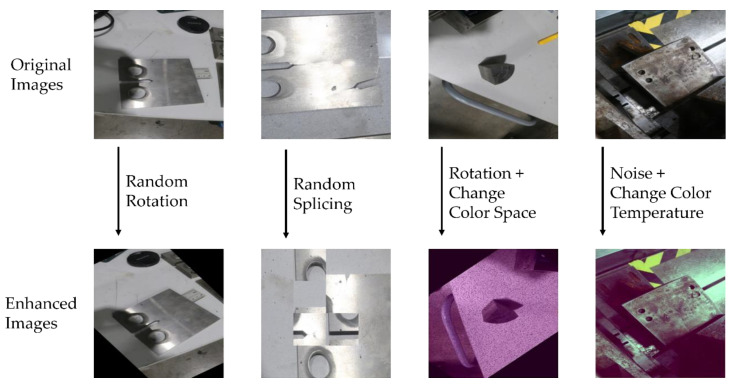
A partial display of the original and enhanced images in the micro-crack dataset.

**Figure 11 sensors-24-00062-f011:**
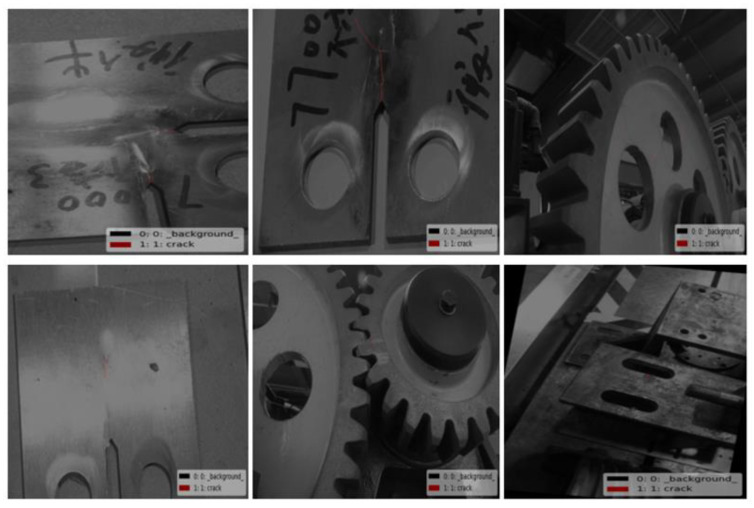
Labeling effect of micro-crack dataset.

**Figure 12 sensors-24-00062-f012:**
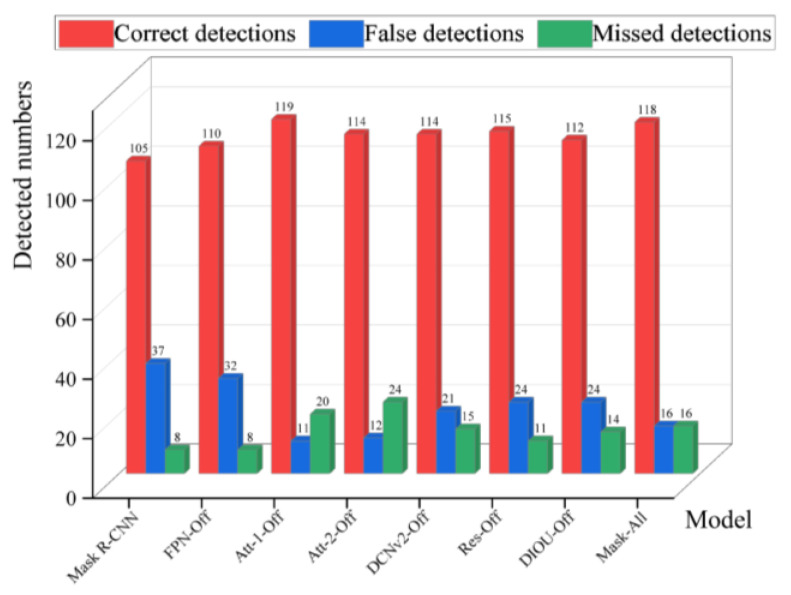
The detection performance of each network.

**Figure 13 sensors-24-00062-f013:**
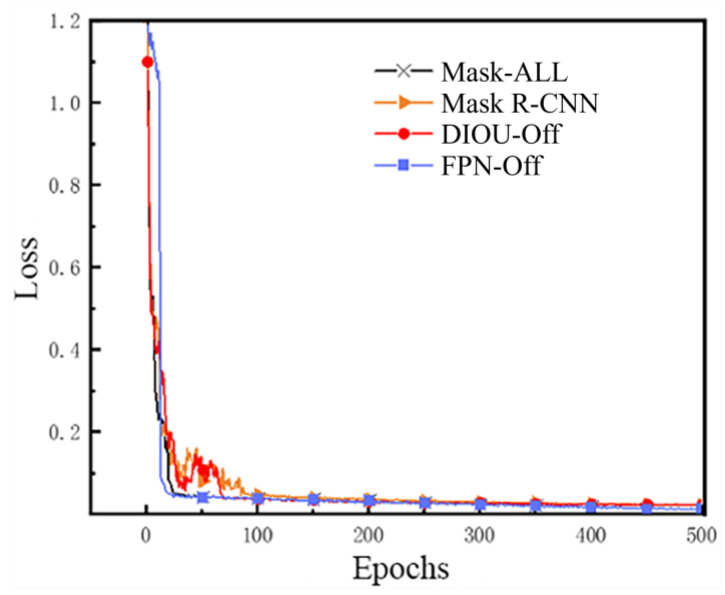
The convergence performance during training of each network.

**Figure 14 sensors-24-00062-f014:**
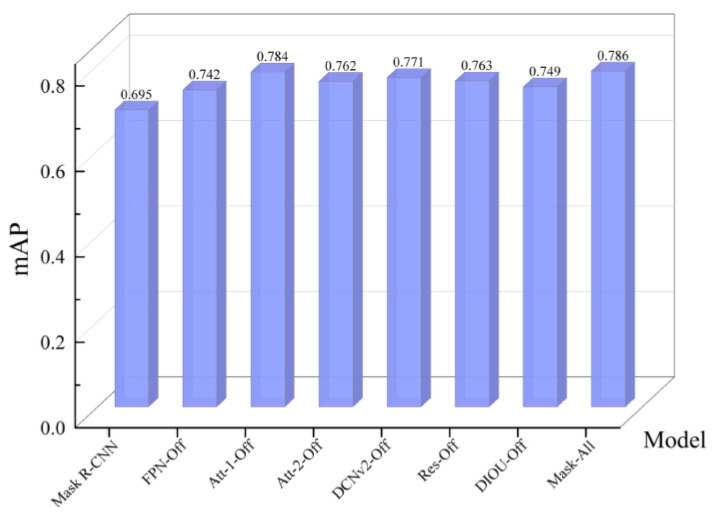
The mAP values of each network.

**Figure 15 sensors-24-00062-f015:**
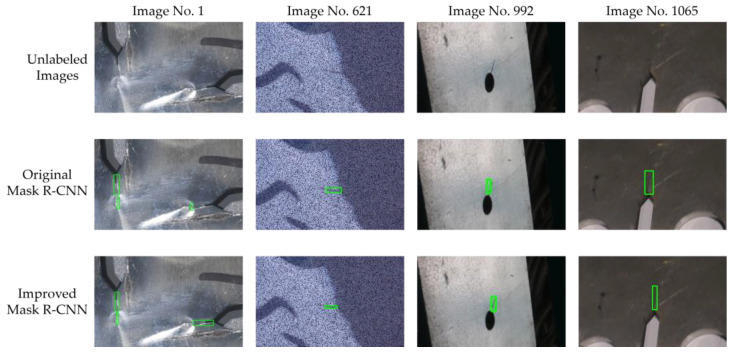
Enlarged display of crack area in four images. The green boxes represent the micro-crack areas identified by models.

**Table 1 sensors-24-00062-t001:** Experimental environment.

Category	Detail Information
CPU	Intel(R) Xeon(R) CPU E5-2609 v4
GPU_1	NVIDIA GeForce RTX 3090
GPU_2	NVIDIA TESLA T4 GPU
RAM_1/GB	100
RAM_2/GB	32
Operating System	Ubuntu 20.04
Operating Environment	Anaconda3, CUDA10.1, Python3.7
IDE	VSCode
Deep Learning Framework	PyTorch
CPU	Intel(R) Xeon(R) CPU E5-2609 v4

**Table 2 sensors-24-00062-t002:** Confusion matrix.

		Predicted Value	
		Positive	Negative
Ground Truth	True	TP	FN
	False	FP	TN

**Table 3 sensors-24-00062-t003:** Hyperparameter value settings of the network model.

Hyperparameter	Epochs	Learning Rate	Batch Size
Value	500	0.001	8

**Table 4 sensors-24-00062-t004:** The position and size of the crack true box.

Image No.	$x_{1}$	$y_{1}$	$x_{2}$	$y_{2}$	Box Length *l*	Box Wight *w*
(1)	801.60	939.00	885.60	954.90	84.00	15.90
	747.30	946.00	799.50	953.30	52.20	7.30
	743.00	1271.40	766.10	1357.30	23.10	85.90
(226)	1094.48	657.51	1215.35	676.93	120.86	19.42
(448)	787.40	559.49	872.35	661.19	84.95	101.69
(465)	971.74	778.31	1021.46	834.37	49.72	56.06
	845.69	858.14	862.88	904.75	17.19	46.61
	1312.30	557.30	1334.00	623.40	21.70	66.10
	1258.40	404.62	1325.72	532.15	67.32	127.53
(1065)	719.44	493.07	731.05	555.24	11.62	62.17
(621)	1068.99	596.76	1104.12	603.97	35.13	7.21
(870)	749.18	968.44	775.74	1034.84	26.56	66.41
(508)	990.12	480.63	1025.06	514.10	34.94	33.47
(946)	888.41	486.20	899.84	528.19	11.42	41.99
(992)	506.19	415.68	513.24	439.06	7.05	23.38

**Table 5 sensors-24-00062-t005:** The position and size of the original Mask R-CNN crack prediction box.

Image No.	$x_{1}^{\prime }$	$y_{1}^{\prime }$	$x_{2}^{\prime }$	$y_{2}^{\prime }$	∆xy	Overall Error
(1)	805.10	935.23	888.31	959.32	14.73%	17.22%
	743.28	949.80	793.25	959.20	38.14%	
	736.88	1272.20	766.98	1259.20	36.36%	
(226)	1095.50	657.62	1215.30	676.02	1.53%	
(448)	782.42	554.58	876.92	665.64	5.11%	
(465)	977.04	782.66	1024.55	838.32	7.92%	
	848.20	853.50	867.56	908.01	14.69%	
	1308.98	553.20	1338.45	626.98	11.86%	
	1257.68	409.21	1329.20	537.50	3.51%	
(1065)	714.87	499.05	736.64	559.79	26.09%	
(621)	1063.95	591.75	1107.11	606.01	30.16%	
(870)	742.85	967.30	778.25	1034.84	8.75%	
(508)	993.01	482.22	1028.99	519.69	10.24%	
(946)	890.23	489.12	896.23	523.98	16.13%	
(992)	503.00	412.83	509.32	434.54	33.09%	

**Table 6 sensors-24-00062-t006:** The position and size of the improved Mask R-CNN crack prediction box.

Image No.	$x_{1}^{\prime }$	$y_{1}^{\prime }$	$x_{2}^{\prime }$	$y_{2}^{\prime }$	∆xy	Overall Error
(1)	803.51	939.00	885.60	954.90	7.01%	7.44%
	747.30	946.00	799.50	953.30	21.21%	
	743.00	1271.40	766.10	1357.30	2.93%	
(226)	1094.48	657.51	1215.35	676.93	0.35%	
(448)	787.40	559.49	872.35	661.19	2.31%	
(465)	971.74	778.31	1021.46	834.37	3.95%	
	845.69	858.14	862.88	904.75	8.28%	
	1312.30	557.30	1334.00	623.40	3.99%	
	1258.40	404.62	1325.72	532.15	2.54%	
(1065)	719.44	493.07	731.05	555.24	15.15%	
(621)	1068.99	596.76	1104.12	603.97	23.84%	
(870)	749.18	968.44	775.74	1034.84	6.33%	
(508)	990.12	480.63	1025.06	514.10	2.99%	
(946)	888.41	486.20	899.84	528.19	3.08%	
(992)	506.19	415.68	513.24	439.06	7.72%	

## Data Availability

The data presented in this study are available on request from the authors.

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
