# Peer review of "An Improved Mask R-CNN Micro-Crack Detection Model for the Surface of Metal Structural Parts"

_sensors, 2023, doi:10.3390/s24010062_

Round 1
Reviewer 1 Report
Comments and Suggestions for Authors
General Comments:
Aiming at the low accuracy of the conventional target detection model during the micro-cracks detecting tasks on the metal structural parts surface, this paper built a micro-cracks dataset, and explored the detection performance optimization method based on Mask R-CNN. The ablation comparison experiments shows that all the improvement schemes proposed in this paper have improved the performance of the original Mask R-CNN. The integration of all the improvement schemes can produce the most significant performance improvement effects in recognition, classification and positioning simultaneously, thus proving the rationality and feasibility of the improved scheme in this paper.
These studies offer valuable insights and contribute to the existing knowledge in this field. The overall organization of the study is commendably well-structured, and it falls within the crack detection of metal structural parts. I have conducted a comprehensive review of the manuscript and would like to commend the authors on the exceptional quality of their work. I recommend the publication of this manuscript in the MDPI's Sensors. However, prior to acceptance, I suggest making some minor revisions to address a few concerns and clarify certain points. These modifications will significantly enhance the clarity and impact of this work. Please find below the specific issues that require attention:
Specific comments:
1. Line 50: What is XGBOOST? Abbreviations should not be used on first occurrence.
2. In section 4.1, more details on the experimental setup and image acquisition of micro-cracks should be provided
3. Lines 506-507: The authors used a high-resolution camera to capture images of micro-cracks. What is the identification result of the proposed method on micro-crack images acquired by low-resolution cameras? Please conduct a comparative analysis.
4. Lines 513-526: The authors collected and obtained the initial dataset of small crack images, which contained 500 images. Then, the image number in the micro-crack dataset is increased from 500 to 2000 by the data enhancement method. Overall, the dataset of micro-crack images is still insufficient may affect the accuracy of also predicting the results.
5. I suggest that the limitations of this work should be discussed in the conclusions.
6. The manuscript is missing several relevant publications in the field, for example:
- Wang S, Liu C, Zhang Y. Fully convolution network architecture for steel-beam crack detection in fast-stitching images. Mechanical Systems and Signal Processing, 2022, 165: 108377. https://doi.org/10.1016/j.ymssp.2021.108377.
- Luo K, Kong X, Zhang J, et al. Computer vision-based bridge inspection and monitoring: A review. Sensors, 2023, 23(18): 7863. https://doi.org/10.3390/s23187863.
- Li K, Li T, Ma M, et al. Laser cladding state recognition and crack defect diagnosis by acoustic emission signal and neural network. Optics & Laser Technology, 2021, 142: 107161. https://doi.org/10.1016/j.optlastec.2021.107161.
Author Response
Thank you for your very valuable comments on our paper, it is very meaningful to us. We have responded to each of your comments. Our response is in the attachment, and please check it.
Thank you for your correction.

Reviewer 2 Report
Comments and Suggestions for Authors
This manuscript presents a Mask R-CNN model to detect micro-cracks on the surface of metal components, which matches the scope of the Journal. It contains some interesting results showing improved detection performance through the integration of several improvement schemes explored in the manuscript. There are however following issues to be addressed before the manuscript can be considered for publication on the journal:
1) The images in Figure 10(a) should be explained properly in terms of representative features in each image and associated challenges to the accurate and reliable detections.
2) It is not clear on the enhancement for the images in Figure 10(b) compared to the original images in Figure 10(a). You could focus on one or two images to demonstrate the enhancement properly.
3) For the images listed in Table 5 and Table 6, the image with the biggest error should be presented and analysed properly. Underlying reasons for the error associated with that particular image should be explored.
4) Lines 168-170 are identical to Lines 163-165. Proofreading should be conducted for the whole manuscript to eliminate this kind of mistakes.
5) ‘this chapter’ on Line 279 should be corrected. This is a manuscript for a journal, not a thesis. Same wording (‘this chapter’) appears several times in the manuscript.
6) What is Dmn in Equation (11)?
7) Second ‘Conv2, Conv3’ in Figure 8 Scheme (b) should be ‘Conv4, Conv5’.
8) ‘However, the disadvantage ….. used for training’ on Lines 472 – 474 doesn’t make sense.
9) Definitions of ‘precision’ and ‘recall’ are the same based on the statements in Lines 547-550. Why?
10) Explain ‘and P and R are inversely related, so if R remains high as P increases, the algorithm is performing well’ (Lines 560-561).
11) ‘figure 14’ on Line 599 is not correct.
Comments on the Quality of English LanguageQuality of English language should be improved. Proofreading should be carried out.
Author Response

(The authors gave the same response as above.)
